# Distribution, dynamics, and physiological races of wheat stem rust (*Puccinia graminis f.sp.* tritici*) on irrigated wheat in the Awash River Basin of Ethiopia

Nurhussein Seid Yesuf[1]*, Sileshi Getahun[1‡], Shiferaw Hassen[1‡], Yoseph Alemayehu[2‡], Kitessa Gutu Danu[3‡], Zemedkun Alemu[1‡], Tsegaab Tesfaye[3‡], Netsanet Bacha Hei[4‡], Gerald Blasch[5‡]

1 Werer Agricultural Research Center, Werer, Afar, Ethiopia, 2 International Wheat and Maize Improvement Center (CIMMYT), Addis Ababa, Ethiopia, 3 Ambo Agricultural Research Center, Ambo, Oromia, Ethiopia, 4 Ethiopian Institute of Agricultural Research, Addis Ababa, Ethiopia, 5 International Maize and Wheat Improvement Center (CIMMYT), Texcoco, Mexico

‡ SG, SH, YA, KGD, ZA, TT, NBH and GB also contributed equally to this work.
* nurhusseinsy@yahoo.com

**Data Availability Statement:** All relevant data are within the manuscript and its Supporting Information files.

## Abstract

Wheat is one of the high-value major crops in the world. However, wheat stem rust is considered one of the determinant threats to wheat production in Ethiopia and the world. So this study was conducted to assess the disease intensity, seasonal distribution dynamics pattern, the genetic variability of *Puccinia graminis* f. sp. *tritici*, and to determine the virulence spectrum in the irrigated ecology of the Awash River Basin. Totally 137 wheat farms were evaluated, from 2014/15–2019/20 in six districts representing the Upper, Middle, and Lower Awash River Basin. Farm plots were assessed, in every 5–10 km intervals, with 'X' fashion, and data on disease incidence, severity, healthy plants were counted and recorded. Diseased samples were collected from the diseased wheat stem by *Puccinia graminis* physiological and genetic race analysis. The seasonal trend of stem rust disease progress showed its importance to infer the future progresses of the disease for the country's potential production plan of irrigated wheat. The result revealed that the disease prevalence, disease incidence, and severity were significantly varied; among the different districts and seasons in the two regions. The survey results also indicated that about 71.7% of the wheat fields were affected by stem rust during the 2018/19 growing period. The disease's overall incidence and mean severity during the same season were 49.02% and 29.27%, respectively. In 2019/20, about 63.7% of the wheat fields were affected by stem rust, disease incidence 30.97%, and severity 17.22% were lower than the previous season. In 2019/20, even though seasonal disease distribution decreased, the spatial distribution was expanding in Afambo and Dubti districts. Four, stem rust dominant races were identified (TTTTF, TKTTF TKKTF, and TTKTF) by physiological and genetic race analysis during 2018/19 and one additional race (TKPTF) in 2019/20, production year. The result indicated that the races are highly virulent and affect most Sr genes except Sr31 and Sr24. From the race analysis result, TTTTF, and TKKTF have the broadest virulence spectrum race, which affects 90% of

**Funding:** DGGW Project for the 2018/19 budget year financial support, and Cambridge University 2019/20 Production year survey.

**Competing interests:** There is no competing of interests.

the Sr genes. Generally, we can conclude that the spatial and seasonal distribution of the disease is expanding. Most of the races in the irrigated areas in the Basin were similar to that of rain-fed wheat production belts in Ethiopia, so care must be given, to effective management of the diseases, in both production ecologies towards controlling the spore pressure than race variability. Therefore, these findings provide inputs for wheat producers to reduce the spread and disease' damage in the irrigated ecologies of Ethiopia. Also, it gives an insight for breeders to think about the breeding program in their crossing lines.

## Introduction

Wheat (*Triticum aestivum* L.) is one of the leading cereal grains where more than one-third of the population uses it as a staple food [1]. As the world population is increasing to reach nine billion by 2050 [2], there should be an urgent need to increase crop productivity like wheat to meet the increasing demand for food [3]. In sub-Sahara Africa (SSA) wheat is grown on a total area of 2.9 M ha with an annual production amounting to 7.5 MT [4]. The usual wheat-producing countries in Sub-Sahara Africa are Ethiopia, South Africa, Sudan, Kenya, Tanzania, Nigeria, Zimbabwe, and Zambia hold the highest benefactions, respectively. Ethiopia accounts for the largest production area (1.7 M ha), followed by South Africa (0.5 M ha) [5]. Wheat production in Ethiopia is expanding at 3.69% from 1979/80-2017/18 [6]. In Ethiopia, the total annual wheat production is 4.64 million tons from 1.7 million hectares of land in the 2018 growing season [7]. Previously, wheat was predominantly produced, in southeastern, central, and northwestern regions of Ethiopia [8], and the government started irrigated wheat production as a mechanism to substitute imports.

Irrigated crop productivity is higher than rain-fed. For decades the Ethiopian government's policies have promoted irrigation expansion, particularly for cereal crops, as a means for improving agricultural growth, smoothing production risk, and alleviating rural poverty, and unlocking food security and safety problems. The production of wheat through irrigation is an attractive business because it is a multi-social responsibility to support the country in food self-sufficiency, source of animal feed for pastoralists, and mechanisms to fill the gap for foreign exchange. However, its production and productivity are low compared with the world average (3.3 t/ha), this could be due to several factors of which biotic (diseases, insects, and weeds), abiotic stress, and low adoption of new agricultural technologies are the major ones [9, 10]. Tadesse *et al.* [5] and Rosegrant *et al.* [11] reported wheat demand was increasing in the world parallel with the challenges in its production like climate change, cost of production inputs increased, and increased abiotic and biotic stresses make the wheat demand-supply chain very volatile. The abiotic production constraints in irrigated lowland areas of Ethiopia are environmental stresses like extreme temperature, soil salinity, drought and flood, soil pH, and salinity [12]. Among the biotic stresses, stems rust caused by *Puccinia graminis* f. sp. *tritici* hear after referred to as *Pgt* could cause 100% yield losses during epidemic years on susceptible cultivars [13, 14].

Stem rust disease was the most prevalent, devastating, and severe menace of all wheat rusts in the country that majorly reduces wheat production [15]. So far stem rust was the common disease in the Awash River Basins except in the Dubti district in the Afar region. The menace could be prominent biotic stress in Ethiopia, with environmental factors related to the other two rust pathogens disease triangle (Fig 2). Surveillance results revealed that the predominance of stem rust in East and southern Africa [16]. In Ethiopia, hot spot areas for the appearance of

virulent genetic diversity of stem rust races were reported by Hailu *et al.* [13]. Recurrent efforts have been made so far by using effective Sr genes in combination with other genes with gene pyramiding and found great importance as the additive effects of several genes offer the cultivar wider base stem rust resistance [17]. Ethiopia's prospect of wheat self-sufficiency in the coming two years will be possible with two favorably realistic cases in addition to positive high yielding promising results. These are increment of production in the rain-fed wheat production areas, and expansion to irrigable high potential untouched lowlands was a strategy to expand production [18]. Recent evidence revealed that irrigation wheat was a portion in increasing wheat production and productivity. The demonstration and expansion of production technologies across the main river basins of the country started from the technological support of the Werer Agricultural research center in the Awash River Basin. In 2018 about 3502 ha, 2019 about 15,100 ha of land in Afar (Amibara, Dubti, and Gewane), Oromia region (Sire and Jeju), South Omo (Arba-Minch zuria, South Omo), and Somalia (Gode) was targeted according to ministry of agriculture [18] and for 2020 about 300, 000 ha of land planned to covered with wheat in irrigation. The area expansion up to 300,000 ha for 2020 production year was in Oromia, Amhara Somalia, Southern Nations and Nationalities and Peoples Regional State, and Afar.

Ethiopia is considered one of the hotspots for new race development, and thus, the new production areas will intensify this theory. The *Pgt* disease could be controlled through growing resistant varieties, knowing the prevalent races, where the required knowledge in the variety development [19]. Stem rust disease prevalence, incidence, and severity parameters with their defined dominant virulent and avirulent races give essential information on the gene-to-gene concept to be considered, in the breeding programs [20]. Among the identified races, TKTTF was dominant at a frequency of 78.7% Hailu *et al.* [21], TKTTF race is the most frequent and has caused a severe epidemic in the south wheat growing regions (Bale and Arsi) after its first detection in 2012 [22].

Our research hypothesis was to address how *Pgt* distributed in irrigated wheat production areas of Awash River Basin, Ethiopia? How is the disease intensity going in the irrigated wheat production areas of the Awash River Basin? What are the likely national scale economic losses from the *Pgt* based upon the field data set? What are the physiological and genetic races of wheat stem rust common in irrigated wheat production districts in the Awash River Basin of Ethiopia? The genetic analysis of isolated *Pgt* races in Cereals Disease Lab (CDL) at the University of Minnesota. New races expected from the new cultivation areas, could there a new race? Generate information on the *Pgt* epidemics dynamics within recurrent expanding production scenarios. The present study targeted to know the distribution of wheat stem rust prevalence, disease incidence, severity, and their virulence spectrum to ensure sustainable irrigated wheat production in Awash River Basin and similar lowland area irrigated wheat production agroecological settings (344–1266 m.a.s.l.).

The objectives of this study were to determine the disease intensity distribution, epidemics dynamics, in the irrigated belts of the Awash River Basin and to analyze the virulence of isolated *Pgt* races on wheat plants for the past six years.

## Materials and methods

### Study site description

The field survey was conducted on upper, middle, and lower Awash River Basins during 2014/15 to 2019/20 irrigated wheat growing seasons. The study includes seven districts having four (Amibara, Afambo& Asaita, Dubti, Samorovia Gellalo, and Gewane) from Afar and three from Oromia regional states (Fentale, Jeju, and Sire). The districts are located in the range between 039˚ 39ʹ 20ʺ E and 41˚43ʹ40" E and 08˚ 25ʹ 50ʺ N and 11˚32'20" N (Fig 1 & S1 Fig).

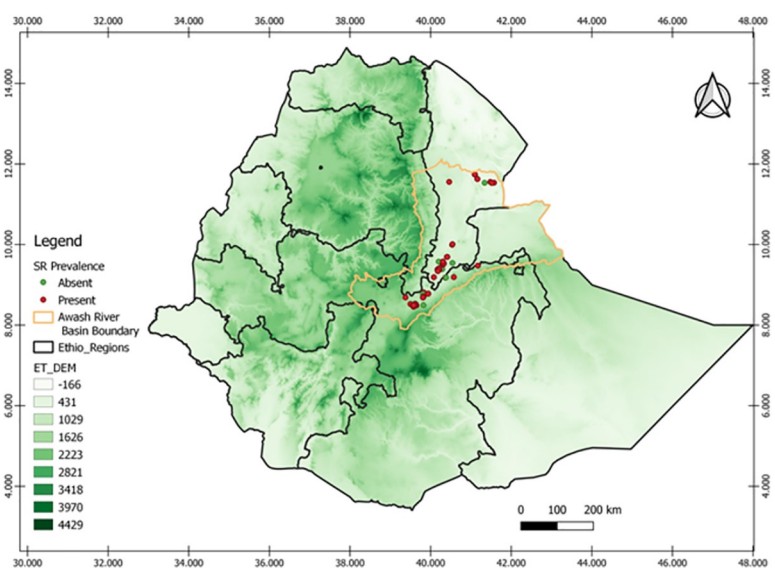

**Fig 1. The irrigated wheat production belt status in Awash River Basin in 2014/15-2019/20 cool cropping season, "reprinted from [ref] under a CC BY 4.0 license, with permission from [PLOS ONE], original copyright [2021]".**

## Ethical statement

The survey was, conducted besides the lateral aim of rust epidemics early warning and monitoring support program in the Awash River Basin. Samples for this study collected from farmers' fields of the irrigated production areas in the Awash River Basin. The disease was an airborne infection that is difficult to contain. For sample collection, no specific permission and conditions were requested, for these locations. Field sites are on public access, and *P. graminis* f. sp. *tritici* is already an air-born pathogen that doesn't need special protection. This work was our study experience in the endeavor of irrigated wheat technology dissemination.

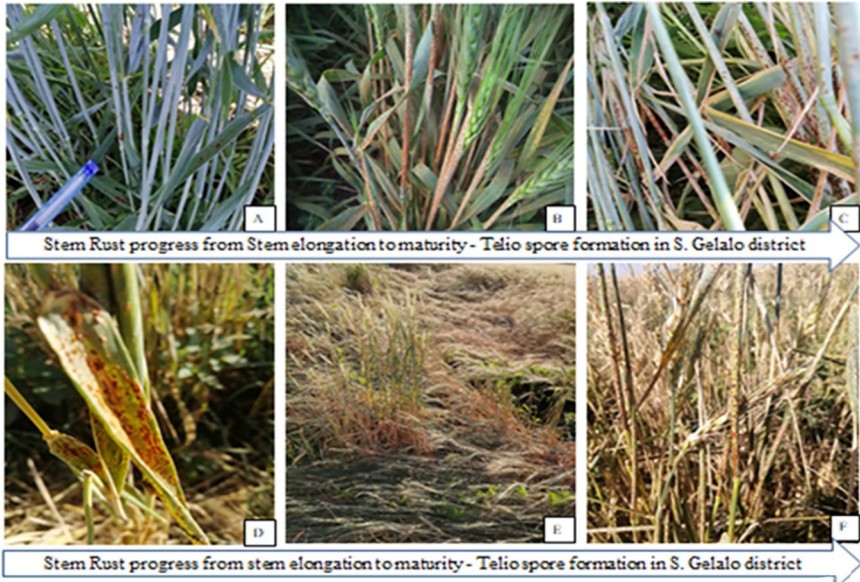

**Fig 2.** Stem rust virulence progress in a different stage of the plant: (A = stem elongation, B = Booting stage, C = milky stage, D = dough stage, E = matured, F = teliospore formed in straw) districts of S. Gellalo districts.

## Survey of wheat stem rust distribution in Awash River Basin fields

The field assessments were conducted in the farmers' fields at 5–10 km interval between wheat farms following the main road routes. Generally, the disease assessments were done in 137 farms over six years in Oromia and Afar regional state of the Awash River Basin. The field visited in each district increased with wheat production adoption expansion over years. The number of farms visited in the Amibara district was three, three, two, three, ten, and nine in 2014/15, 2015/16, 2016/17, 2017/18, 2018/19, and 2019/20 offseason wheat production years, respectively. The number of fields visited in Afambo and Asaita districts was five in the 2019/20 production year. The number of farms visited in the Dubti district was five and four in 2018/19 and 2019/20 on offseason wheat production, respectively. The number of fields visited in the Samorovia Gellalo district was four in 2018/19 on the offseason wheat production year. The number of farms visited in the Gewane district was one, three, and one in 2015/16, 2017/18, and 2018/19 on the offseason wheat production years, respectively. The number of farms visited in the Fentale district were seven, seven, eight, five, ten, and 17 in 2014/15, 2015/16, 2016/17, 2017/18, 2018/19, and 2019/20 production years, respectively. The number of farms visited in the Jeju district was ten and six in 2018/19 and 2019/20 production years, respectively. The number of farms visited in Sire, district six, and five in 2018/19 and 2019/20 production years, respectively (Figs 1 and 6).

The assessment was done, in the double diagonal pattern, and samples taken using a 1m$^2$ quadrate. In each wheat field, plants within the quadrate were counted and recorded as infected and healthy. Stem rust disease intensity was calculated, as disease prevalence, disease incidence, and severity with the respected formula given below. Usually, for the management initiation, a cut score to classify afield as diseased was 5% incidence score; but for this study, we scored and registered the disease prevalence, incidence, and the disease severity on field condition recorded using the modified Cobb's scale as developed by Peterson et al. [23]. The observed farms during the assessment were in the crop stage range of booting to physiological maturity.

$$\textbf{Prevalence}: \textbf{P}(\%) = \left( \frac{\textbf{Number of fields affected}}{\textbf{Total number of fields inspected}} \right) * \textbf{100}$$

$$\textbf{Disease Incidence}: \textbf{DI}(\%) = \left( \frac{\textbf{Number of diseased plants in aquadrant}}{\textbf{Total number of plants in aquadrant}} \right) * \textbf{100}$$

$$\textbf{Disease Severity}: \textbf{DS}(\%) = \left( \frac{\textbf{Area of aplant tissue affected}}{\textbf{Total area of aplant part assessed}} \right) * \textbf{100}$$

## Seasonality and spatial epidemics dynamics

Data on disease intensity like prevalence, disease incidence, the severity of the disease, and the prevalent physiological races virulence spectrum were compared to see the seasonal variations and space in the Awash River Basin irrigated wheat. The virulence spectrum/avirulence assessment was made to the identified races using the 20 stem rust resistance, differential lines genes.

## Physiological race identification of *P. graminis* f. sp. *Tritici*

**Collection of wheat stem rust samples.** Samples of infected stems on diseased wheat fields and trial plots were collected in Awash River Basin Oromia and Afar regional state of Ethiopia, to the period 2018/19-2019/20. Stems infected with wheat stem rusts were cut into small pieces of 5–10 cm using scissors and put in paper bags after the leaf sheath was separated from the stem to keep the stem rust viable and leaf sheath dry. During sample collection, we used absolute alcohol to sterilize the scissors to avoid cross-contamination among isolates. Samples collected in the paper bags were tagged, with the name of the zone, district, longitude, latitude, variety, sample collection date, and transported to Ambo Agricultural Research Centers (AARC) for race analysis. The air-dried samples kept within paper bags were kept in a refrigerator at 4°C until the survey completed in all the study districts. Five seedlings were raised, for a McNair variety, in each sample in suitable 8cm diameter clay pots, filled with a mixture of steam-sterilized soil, sand, and manure in the ratio of 2:1:1, respectively. Spores were collected using the capsule from stem rust infected samples using a collector. The collected urediniospores from fields were suspended, in lightweight mineral oil soltrol-170. Then inoculated using atomized inoculator, on 7-day-old seedlings, variety McNair, which does not carry known stem rust resistance genes, to collect enough spores inoculate the stem rust differentials [24]. When the pustules developed at two weeks, spores from each pustule were collected using a power-operated vacuum aspirator and stored separately in gelatin capsules. A suspension prepared to mix urediospores with Soltrol 170 oil was inoculated on seven-day-old seedlings of the susceptible variety McNair for multiplication purpose for each of a single pustule, on separate pots following the procedure mentioned above. The urediniospores descending from one pustule made up a single pustule isolate. One isolate was developed from each wheat field and used for the final race analysis [24].

Greenhouse inoculations were done using the methods and procedures developed by Stakman *et al.* [25]. The mono pustule inoculated plants were then moistened, with fine droplets of distilled water, produced with an atomizer, and placed in dew chamber for 18h dark at 18 to 22°C followed by exposure to light for 3 to 4h to provide the condition for infection and seedlings were allowed to dry for about 2hours. Then seedlings transferred from the dew chamber to the growth room in the greenhouse where conditions regulated at 12h photoperiod at a temperature of 18 to 25°C and relative humidity of 60 to 70%.

Inoculated seedlings were moistened with fine droplets of distilled water, produced with an atomizer, and placed in a dew chamber in darkness for 18 h at 18 to 22°C and 98 to 100% relative humidity. Upon removal from the growth chamber, plants were exposed to 4 hours of fluorescent light to provide a condition for infection and allowed to dry their dew for about 2 hours. The inoculated plants were transferred, into greenhouse benches, condition regulated at 12h photoperiod, 18–25°C temperature, and 60–70% relative humidity [26]. Seven to ten days after disease inoculations, the fleck's or chlorosis were visible on leaves with a single fleck that produces a single pustule or uredinal isolate. A single pustule sample selected from leaves and the remaining seedling within the pots were removed, by scissors. Only leaves with a single pustule from each location were separately covered with Cellophane bags and tied up at the base with a rubber band to avoid cross-contamination [27].

**Inoculation of differential lines and race determination.** Five seeds for each of the twenty wheat stem rust differentials with known stem rust resistance genes (Sr5, Sr6, Sr7b, Sr8a, Sr9a, Sr9b, Sr9d, Sr9e, Sr9g, Sr10, Sr11, Sr17, Sr21, Sr24, Sr30, Sr31, Sr36, Sr38, SrTmp, SrMcN, and a susceptible variety McNair) were grown in 10cm diameter pots. The variety 'McNair' without Sr gene, was used to ascertain the viability of spores inoculated to the differential hosts. Each rust isolate derived from a single pustule was suspended, in Soltrol-130. The suspension was adjusted to 3–5 mg urediospores with 1ml of mineral oil (soltrol-130) and inoculated onto differentials. After disease inoculation, plants were moistened, with fine droplets of distilled water, produced with an atomizer, and placed in an incubation chamber for an 18 hours dark period at 18–22°C and 3–4 hours of light. Upon removal from the dew chamber, plants were placed in separate glass compartments in a greenhouse to avoid contamination and produce infection. Greenhouse temperature was maintained, between 18°C and 25°C. Natural daylight was supplemented, for additional 4 hours/day with 120 µ $E.M^{-2} S^{-1}$ photosynthetically active radiations emitted by cool white fluorescent tubes arranged directly above plants to favor the pathogen produce infection.

Stem rust infection types (ITs) on the differential lines were scored 14 days after inoculation using a 0 to 4 scale [25]. Five letter nomenclature systems are used, for coding the races, [19, 28, 29]. In this study, the differential lines are grouped into five sub-sets, as shown in Table 1, in the following order:

i. *Sr5, Sr2l, Sr9e, Sr7b*

ii. *Sr11, Sr6, Sr8a, Sr9g,*

**Table 1. List of twenty wheat differential hosts for stem rust with their corresponding Sr genes and origin/pedigree.**

| Differential hosts | Sr genes | Origin/Pedigree |
|---|---|---|
| LcSr24Ag | 24 | Little Club/Agent (Cl 13523) |
| W2691SrTt-1 | 36 | Cl12632 T. timopheevii |
| ISr7b-Ra | 7b | Hope/Chinese Spring |
| ISr8a-Ra | 8a | Rieti/Wilhelmina//Akagomughi |
| CnSSrTmp | Tmp | Triumph 64(Cl 13679)/ Chinese Spring |
| Sr31(Benno)/ 6*LMPG | 31 | Kavkaz |
| CnS-T-.mono-deriv | 21 | Einkorn Cl 2433 |
| Trident | 38 | Spear*4/VPM (Pl519303) |
| ISr9a-Ra | 9a | Red Egyptian/Chinese Spring |
| ISr9d-Ra | 9d | Hope/Chinese Spring |
| Combination VII | 17 | Esp 518/9 |
| ISr5-Ra | 5 | Thatcher/Chinese Spring |
| ISr6-Ra | 6 | Red Egyptian/Chinese Spring |
| W2691Sr9b | 9b | Kenya 117A |
| Vernsteine | 9e | Little Club//3*Gabo/2* |
| W2691Sr10 | 10 | Marquis*4/Egypt A95/2/2*W2691 |
| BtSr30Wst | 30 | Festival/Uruguay C10837 |
| CnsSr9g | 9g | Selection from Kubanka(Cl1516) |
| ISr11-Ra | 11 | Kenya C6402/Pusa4/Dundee |
| McNair 701 | McN | Cl 15288 |

Source; Ambo plant protection research center, 2009

**Table 2. Wheat stem rust differentials and nomenclature of *Pgt* based on 20** ** **wheat hosts.**

| *Pgt* Code | Infection types produced on near-isogenic Sr-Gene (Differential hosts) | | | |
|---|---|---|---|---|
| | Set 1 | 5 (*ISe5-Ra*) | 21 (*Cns_T_mono*) | 9e (*Vernstine*) | 7b (*ISr7b-Ra*) |
| | Set 2 | 11 (*Isr11-Ra*) | 6 (*ISr-6-Ra*) | 8a (*ISr8a-Ra*) | 9g (*CnSr9g*) |
| | Set 3 | 36 (*W2691SrTt-1*) | 9b (*W2691Sr9b*) | 30 (*BtSr30Wwst*) | 17 (*Combination V*) |
| | Set 4 | 9a (*ISr9a-Ra*) | 9d (*ISr9d-Rd*) | 10 (*W2691Sr10*) | Tmp (*CnsSrTmp*) |
| | Set 5 | 24 (*LeSr24Ag*) | 31(*Sr31/6*LMPG*) | 38 (*VPM1*) | McN (*McNaire701*) |
| B | | Low [a] | Low | Low | Low |
| C | | Low | Low | Low | High [b] |
| D | | Low | Low | High | Low |
| F | | Low | High | High | High |
| G | | Low | High | Low | Low |
| H | | Low | High | Low | High |
| J | | Low | High | High | Low |
| K | | Low | High | High | High |
| L | | High | Low | Low | Low |
| M | | High | Low | Low | High |
| N | | High | Low | High | Low |
| P | | High | Low | High | High |
| Q | | High | High | Low | Low |
| R | | High | High | Low | High |
| S | | High | High | High | Low |
| T | | High | High | High | High |

Source; Rolefs and martens (1988); Jin *et al*., 2008.

[a]Low = infection types 0,; , 1, and 2 and combinations of these values.

[b] High = infection types 3 and 4 and a combination of these values

 iii. *Sr36, Sr9b, Sr30, Sr17*.

 iv. *Sr9a, Sr9d. Sr10, SrTmp*,

 v. *Sr24, Sr31, Sr38, SrMcN* (Table 1).

An isolate that produces a 'low' infection type on the four lines in a set is assigned, with the letter 'B' in comparison, a 'high' infection type on the four lines is assigned, with the 'T' letter. Hence, if an isolate produces a low infection type (resistant reaction) on the 20 differential lines, the race could be designated with a five-letter race code 'BBBBB' while an isolate producing a high infection type race could be designated, with a five-letter race code 'TTTTT' and race analysis was by using the descriptive statics referred as in in Table 2.

## Dead stem rust samples for genetic race analysis

Stem rust, genetic race analysis, was done to sample sets collected in 2019. A total of 9 samples were collected from widely dispersed and representative locations in irrigated wheat-growing areas. The isolate was collected, by cutting immediately above and below a single pustule using sharp scissors. Sheath tissue was cut, at the side, and the stem inside was removed. Then samples were kept, in 80% alcohol for seven days. After seven days by pouring the alcohol the samples were left open to dry for two days. Then the samples were sealed tubes shipped to Cereals Disease Lab (CDL) Minnesota for DNA analysis.

## Data analysis

Disease intensity parameters like prevalence, disease incidence, and severity were analyzed using descriptive statistical analysis (means) over districts, varieties, altitude range, and crop growth stages. Similarly, race analysis was analyzed using those descriptive statistics [30].

## Results

### Distribution of stem rust invasion in Awash River Basin

During 2014/15, 2015/16, and 2017/18, *Pgt* was intercepted, in single farm spots, and 2016/17, there was no detection of stem rust epidemics at all studied locations. This means the diseases stem rust was not economical in those seasons on wheat variety Gambo. The above scenario could become possible, due to the production area were new and there is no source of inoculum for disease outbreak, or maybe those production seasons were not conducive for rust outbreak. In 2018/19 total of 33 stem rust live samples were taken, and only (12.12%) of samples were viable, while in 2019/20 total of 46 stem rust live samples were taken, and (71.74%) of samples were viable. In the two production seasons, 79 stem rust samples were collected and 46.84% of the sample's remained viable. The spatial viability of the stem rust samples collected in 2019/20 in Oromia Regional State was 40% in Sire, 100% in Jeju, and 70.59% in Fentale districts, while in Afar Regional State the viability was 33.3% in Afambo, 100% in Amibara and 25% in Dubti districts were viable referred as in Table 3.

In 2014/15, the number of fields assessed was three in Amibara and seven in Fentale district a farm was affected with stem rust at physiological maturity in Fentale district Gedara Kebele. Even though the race analysis was lucking, the first interception of the disease was in Fentale district Gedara Kebele in 2014/15 production year with prevalence (14.3), incidence (1.4), and severity (0.7) in percentage. In 2015/16, the stem rust expansion increased to a disease prevalence of 20%, 1% incidence, and 1.5% severity in the Fentale district showed in Table 3. During the same year, stem rust infestation coincided with the late dough stage, such that the diseases were visible in ditches of undulated sites shown in Table 3.

In 2016/17, stem rust pressure was not prevalent in the Awash River Basin irrigated wheat valley. This could be the stem rust disease growth was inhibited by the sunny weather conditions from 15 December to 15 January. There could be several factors in the disease triangle attached to it.

In 2017/18, the stem rust started to invade the wheat production in Amibara with disease incidence (1.33%), severity (0.5%), while there was no disease detection or expansion in the Fentale district showed in Table 3.

In 2018/19, the diseases escalated in terms of prevalence (71.7%), disease incidence (47.45%), and severity (29.27%) in the Awash River Basin. The highest disease incidence of stem rust was recorded in Gellalo (83.33%), the second-largest disease incidence was in Fentale (65%), and the third disease incidence was in Amibara (60%). In the Sire and Jeju districts of the Oromia region, the disease incidence of *Pgt* was (30%) and (50%), respectively, whereas the lowest disease incidence was in Dubti (;) during physiological maturity. The disease severity showed a similar trend as the disease incidence. The highest severity in Gellalo with range mean values 60–80%, (70%) respectively. The disease incidence in Fentale and Amibara districts ranges from 0–80%, and the mean value of (40%) in Fentale and (35.63%) in the Amibara district showed in Table 3.

In Sire and Jeju districts, the severity was (16.67%) and (28.5%) respectively stated in Table 3. Gewane district was rust-free due to poor management, and the performance of the wheat stand was also poor due to improper crop management after booting, which stopped

**Table 3. Prevalence, incidence, and severity of *Pgt* in the districts of Awash River Basin in 2014/15-2019/20 production years.**

| Year | Region | District | Altitude range (m) | NFI | P (%) | I (%) | | S (%) | |
|---|---|---|---|---|---|---|---|---|---|
| | | | | | | Range | Mean | Range | Mean |
| 2014/15 | Afar | Amibara | 735–742 | 3 | 0 | - | - | - | - |
| | Oromia | Fentale | 1030–1045 | 7 | 14.3 | 0–10 | 1.4 | 0–5 | 0.7 |
| | Total | | 735–1045 | 10 | 10 | 0–10 | 1.0 | 0–5 | 0.5 |
| 2015/16 | Afar | Amibara | 739–742 | 3 | 0 | - | - | - | - |
| | | Gewane | 570 | 1 | 0 | - | - | - | - |
| | Oromia | Fentale | 1096–1108 | 10 | 20 | 0–5 | 1.0 | 0–10 | 1.5 |
| | Total | | 570–1108 | 14 | 14.3 | 0–5 | 0.71 | 0–10 | 1.07 |
| 2016/17 | Afar | Amibara | 741–742 | 2 | 0 | - | - | - | - |
| | Oromia | Fentale | 1090–1104 | 8 | 0 | - | - | - | - |
| | Total | | 741–1104 | 10 | 0 | - | - | - | - |
| 2017/18 | Afar | Amibara | 740–742 | 3 | 33.3 | 0–20 | 6.6 | 0–10 | 3.3 |
| | | Dubti | 349–373 | 3 | 0 | - | - | - | - |
| | Oromia | Fentale | 995–1103 | 5 | 0 | - | - | - | - |
| | Total | | 349–1103 | 11 | 9.09 | 0–20 | 1.82 | 0–10 | 0.91 |
| 2018/19 | Afar | Amibara | 740–742 | 10 | 80 | 0–100 | 60 | 0–80 | 35.63 |
| | | Gewane | 570 | 1 | 0 | - | - | - | - |
| | | S. Gellalo | 555–565 | 4 | 100 | 50–100 | 81.25 | 40–80 | 60 |
| | | Dubti | 354–365 | 5 | 0 | - | - | - | - |
| | Oromia | Fentale | 999–1108 | 10 | 80 | 0–100 | 65 | 0–80 | 40 |
| | | Sire | 1262–1266 | 6 | 66.7 | 0–80 | 30 | 0–55 | 16.67 |
| | | Jeju | 1223–1256 | 10 | 90 | 0–90 | 50 | 0–40 | 28.5 |
| | Total | | 349–1266 | 46 | 71.74 | 0–100 | 49.02 | 0–80 | 29.27 |
| 2019/20 | Afar | Amibara | 740–742 | 9 | 37.5 | 0–100 | 17.5 | 0–80 | 9.38 |
| | | Afambo&Asaita | 344–365 | 5 | 20 | 0–10 | 2 | 0–5 | ; |
| | | Dubti | 354–365 | 4 | 25 | 0–75 | 18.75 | 0–35 | 8.75 |
| | Oromia | Fentale | 995–1119 | 17 | 88 | 0–100 | 49.12 | 0–80 | 30.88 |
| | | Sire | 1262–1266 | 5 | 60 | 0–80 | 37 | 0–50 | 18 |
| | | Jeju | 1223–1256 | 6 | 100 | 0–70 | 22 | 0–20 | 10 |
| | Total | | | 46 | 63.7 | 0–100 | 30.97 | 0–80 | 17.22 |

NFI = No. diseased field inspected; P = prevalence I = incidence; and S = severity

irrigation water at the booting stage completely. Even though the late planting date increases the outbreak of stem rust diseases with its polycyclic spore formation and the air had born transmission nature. Stem rust is polycyclic by its nature, this character aggravated with the movement of air current or air-assisted expansion, spore pressure in the air was critical to the nature of damage and economics of in the disease virulence. Most of the varieties released in Werer agricultural research centers, including the Fentale 2, were susceptible to stem rust. Fentale two ranges with 100% incidence and 80% severity in Werer Seed multiplication leased farm. Stem rust disease also affected the seed multiplication unit of the center and caused high yield losses. In 2018/19, the disease invades most wheat production districts in the basin and causes economic losses in terms of agricultural input cost for fungicide purchases, application cost, and yield losses in (Fig 2) and Table 3. The disease pressure was severe in farms near Awash River with high morning dew pressure and low on farms far from riverbanks. The disease pressure was severe in the farms near Awash River. The disease pressure follows the

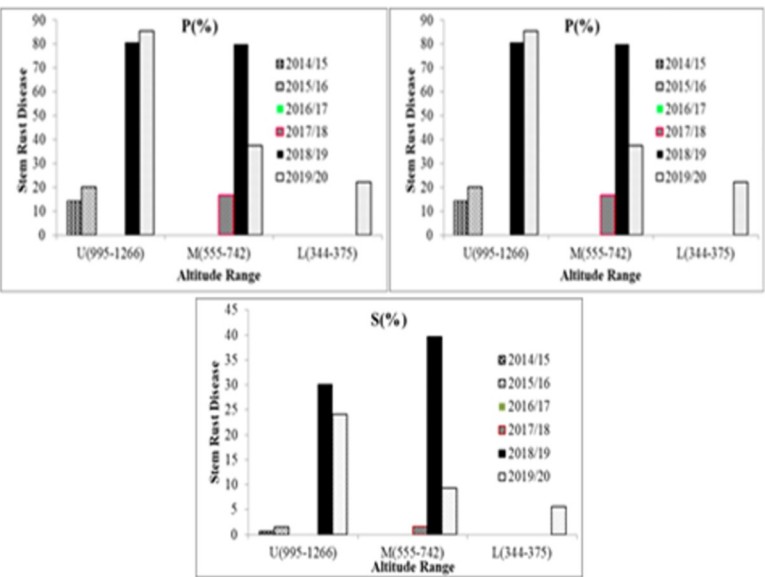

**Fig 3.** A (P%) = Prevalence, B (I%) = Disease incidence, and C (S%) = Severity percentages of stem rust disease in with three altitude range of Awash Basin areas considered of; U = Upper Awash, M = Middle Awash, and L = Lower Awash. "Reprinted from [ref] under a CC BY 4.0 license, with permission from [PLOS ONE], original copyright [2021]".

morning dew weight gradient with high, in farms at the edge of river banks and low on farms far from riverbanks.

In 2014/15, the prevalence and disease intensity of stem rust in small-scale producers' fields were negligible in the Awash River Basin; the first intercept in irrigated wheat production in Ethiopia was in the Fentale district for the year. The most probable reason for low disease intensity was a low source of inoculum or the absence of the pathogen in the new production area. In 2015/16, Wheat production was at small scale farmers of Amibara, Gewane, and Fentale districts, the disease prevalence and intensity of stem rust in small scale producers in Awash River Basin fields were still low; with an increment of the assessment to 10 fields prevalence in Fentale districts from 14.3% to 20.0% the intensity was still negligible. In 2017/18, there was no stem rust disease prevalence in irrigated lowlands. In 2018/19, wheat production was expanded, to large-scale producers and sugar corporation farms. The stem rust prevalence reached 80% in Amibara, 100% in the S.Gellalo districts of Middle Awash, and there was no *Pgt* infection in Lower Awash, also known as Tendaho sugar corporation farms at Dubti. While the prevalence of the diseases was 80% in Fentale, 90% in Jeju, and 66.7% in Sire districts of Oromia. In 2019/20, wheat production grows exponentially and reached 20,000 ha, and the disease prevalence was 100% in Jeju, 88% in Fentale 60% in Sire districts of Oromia. In particular, the prevalence was relatively lower in the lowlands of Afar. There was 37.5% in Amibara, 25% in Dubti, and 20% in Afambo and Asaita showed above in Table 3.

## Stem rust disease dynamics in Awash River Basin

*Puccinia graminis* f.sp *tritici*, dynamics in the low land areas its disease measuring parameters and infection increased from production year to year. The *Pgt* disease prevalence shown above in Fig 3A was increased, from time to time with the expansion of wheat production and reached 80% prevalence in the 2018/19 production year on Upper Awash and Middle Awash areas. In 2019/20, the disease prevalence of wheat stem rust was more prevalent in Upper Awash which was 85.6%, the disease prevalence was 30% in Middle Awash, and 20%

prevalence in Lower Awash, in farmers and Sugar Corporation Seed Multiplication fields showed in (Fig 3).

*Pgt* disease intensity was highest in Middle Awash than Upper Awash and Lower Awash which, is healthy for 2018/19. The disease incidence of *Pgt*, as depicted above in Fig 3B, was increasing from year to year. The result indicated that *Pgt* disease intensity, in the Awash River Basin was majorly influenced by weather factors. In 2018/19, the weather condition of the Awash River Basin area was cloudy, hot, and humid which is conducive for stem rust epidemics. *Pgt* incidence in 2018/19 was higher in Middle Awash than in Upper Awash. Generally speaking, the disease pressures were higher on the Upper and Middle Awash than Lower Awash (Figs 3 and 4). That could be possibly related to the environmental condition in Lower Awash, i.e., hot sunny with high evapotranspiration, which removes the dews before germination of spores to the occurrences of disease infection.

In 2018/19, out of the 46 fields, 56.5% of the fields surveyed were found at Upper Awash, with an altitude range of (995–1266) m.a.s.l., 32.61% of the fields, were at Middle Awash, with an altitude range of (555–742) m.a.s.l., and the remaining 10.87% was in Lower Awash at an altitude range of (344–375) m.a.s.l., in Awash River Basin classification. The data showed that the number of stem rust-infected fields decreased as the altitude decreased. The disease incidence also decreased, from 51.15% at Upper Awash to 61.66% at Middle Awash and 0% at Lower Awash. Regarding the disease severity distribution, 32.76% was at Upper Awash, 39.75% was at Middle Awash, and 0% was at Lower Awash fields.

In 2019/20 of the 46 field's, 60.86% of the field's surveyed were found, at an altitude range of (995–1266) m.a.s.l., in upper awash, 19.57% of the fields surveyed were at an altitude range of (555–742) m.a.s.l., in Middle Awash, 19.57% of field's were at altitude range (344–375) m.a.s.l. and in lower Awash in Awash Basin classification.

The data showed that the number of stem rust-infected fields decreased as the altitude decreased. The disease incidence also decreased, from 41.14% at Upper Awash to 17.5% at Middle Awash and 8.33% at Lower Awash in the Awash River Basin. The same result was recorded regarding the disease severity 24.11% at Upper Awash, 9.38% at middle awash, and

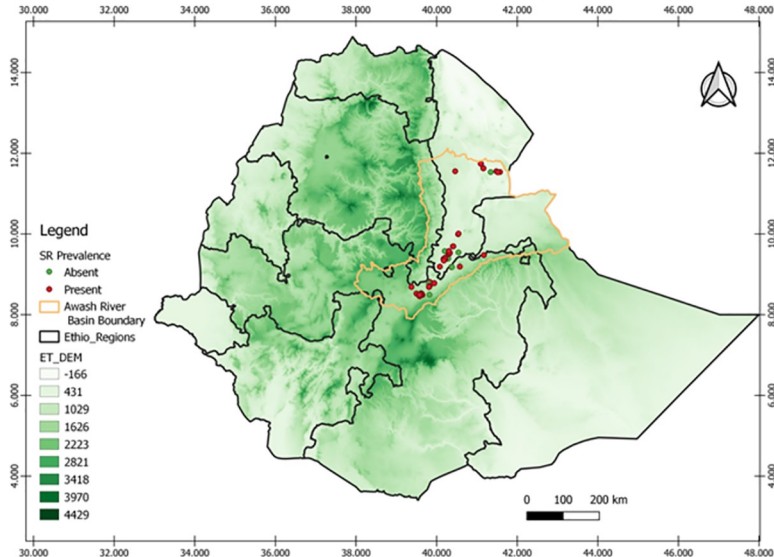

**Fig 4. The effect of attitude on the *Pgt* disease distribution the ET-DEM shows the altitudinal gradients the greener the highland area "reprinted from [ref] under a CC BY 4.0 license, with permission from [PLOS ONE], original copyright [2021]".**

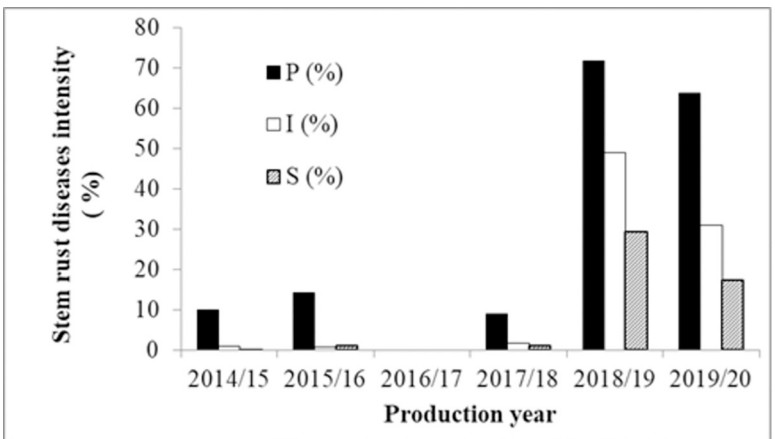

**Fig 5. Prevalence, incidence, and severity of stem rust dynamics year to year of the areas considered.**

3.89%, which is a trace in Lower Awash fields. In the rain-fed agroecology from Ethiopia during the hot season, Ayele *et al.* [31] reported that the highest level of *Pgt*, infection was recorded, in the altitude range of 1600–2500. High humidity, along with high temperatures, during the growing season in the low altitude areas favors the growth of the pathogen [24]. In cool-season, irrigated lowland wheat production, wheat stem rust was important in altitude range above 555 m.a.s.l., Middle and Upper Awash areas. Our result strengthens the findings of the above authors and the environmental requirement of rusts determined by the microclimate condition than the blanket elevation. Generally, the stem rust disease intensity was higher in the Middle, and Upper Awash River Basin, and the initial infection was started in Lower Awash in 2019/20 as depicted in the graph below (Fig 5).

Stem rust diseases in the Awash River Basin were insignificant and low like its production up to 2017/18. The disease infection as prevalence and intensity in the 2018/19 and 2019/20 production years was increased and reached epidemic levels, as the progress of new area production coverage was expanding (Figs 5 and 6).

## Virulence and physiological races of stem rust pathogen

Distribution and frequency of *Pgt* races in Awash River Basin was started in 2018/19 production year. In 2014/15–2017/18 irrigated wheat cropping seasons there was no race analysis, as the disease was not economical, during these production years. In 2016/17, there was no stem rust detection in the Awash River Basin, which could be resulted' from sunny weather conditions during the vulnerable stage of the host plants in the area. In 2018/19, 33 stem rust live samples were taken to the Ambo Agricultural Research Center stem rust laboratory. Only (12.12%) of the samples were viable, which are of different races due to unknown reasons related to the late arrival of the viability of the collected samples was low Table 4. In 2019/20, 46 stem rust survey samples were collected and taken to AARC for physiological race analysis. During this cropping season, a total of 28 stem rust samples in Oromia (Sire, Jeju, and Fentale) districts and 19 samples were collected from Afar regional state (Afambo, Dubti, and Amibara) districts. In 2019/20, 46 stem rust samples were collected from Awash River Basin, and (71.74%) of the samples were viable. The rest (28.26%) of the collected samples were not viable, due to unknown reasons, which are possible in many cases with mistakes during sampling, the gap in the field history like fungicide sprayed samples and sample handlings. Among the collected samples in the Oromia region, 100%, 70.59%, and 40% were viable in Jeju, Fentale, and

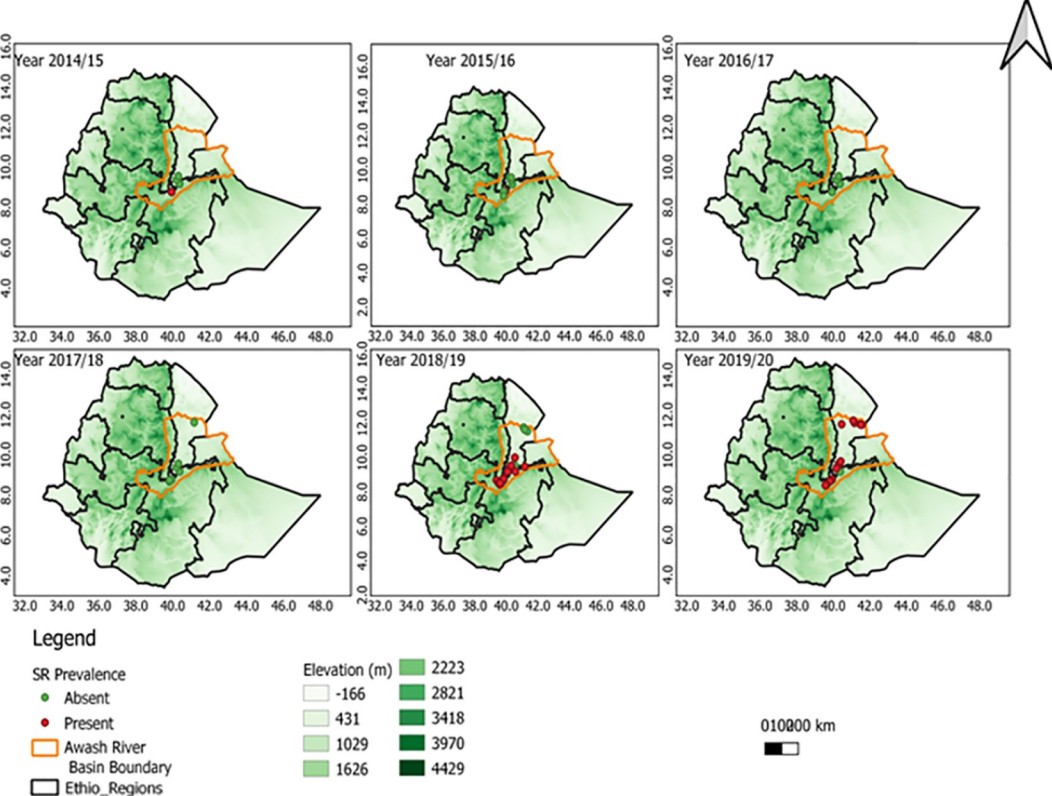

**Fig 6. The Stem rust diseases progress in Awash River Basin over the six years (2014/15-2019/20) "reprinted from [ref] under a CC BY 4.0 license, with permission from [PLOS ONE], original copyright [2021]".**

Sire districts, respectively. While in the Afar region, 100%, 33.33%, and 25% were viable in Amibara, Afamboand Asaita, and Dubti districts, respectively Table 4. Urediniospores, collected from each field were suspended in lightweight mineral oil called Soltrol-170 and

**Table 4. Distribution and frequency of P.gt. races in Awash River Basin irrigated wheat in 2018/19 and 2019/20 for the irrigated wheat cropping season.**

| Races | Year | Afar | | | Oromia | | | Total (%) |
|---|---|---|---|---|---|---|---|---|
| | | Gebiresu | Zone #1 | | East Shewa | East Arsi | | |
| | | Amibara (%) | Dubti (%) | Afambo (%) | Fentale (%) | Sire (%) | Jeju (%) | |
| TTTTF | 2018/19 | 1(100) | - | - | - | - | - | 1(25) |
| | 2019/20 | 5(62.5) | - | - | 5 (41.7) | 2(66.6) | 3(42.9) | 15(45.5) |
| TKTTF | 2018/19 | - | - | - | 1(50) | - | - | 1(25) |
| | 2019/20 | 2(25) | - | - | 3 (25) | - | - | 5(15.2) |
| TKKTF | 2018/19 | - | - | - | - | - | 1(100) | 1(25) |
| | 2019/20 | 1(12.5) | 1(100) | 2(100) | 2 (16.7) | - | 2(28.6) | 8(24.2) |
| TTKTF | 2018/19 | - | - | - | 1(50) | - | - | 1(25) |
| | 2019/20 | - | - | - | 1 (8.3) | 1(33.3) | 2(28.6) | 4(12.1) |
| TKPTF | 2019/20 | - | - | - | 1 (8.3) | - | - | 1(3.0) |
| Total | 2018/19 | 1 | 0 | 0 | 2 | 1 | 0 | 4 |
| | 2019/20 | 8 | 1 | 2 | 12 | 3 | 7 | 33 |

[1]Figures in parenthesis indicated the frequency (%) of a specific race in respective districts, (-) indicated the race was not present in the districts, four samples out of 33 (samples) in 2018/19 and33 (46) (samples)

**Table 5. The *Pgt* races climatic distribution in 2018/19 and 2019/20 cool seasons irrigated wheat areas.**

| Races | Upper Awash (995–1266) | | Middle Awash (555–742) | | Lower Awash (344–375) | |
|---|---|---|---|---|---|---|
| | 2018/19 | 2019/20 | 2018/19 | 2019/20 | 2018/19 | 2019/20 |
| TTTTF | - | 10 (45.5) | 1 (100) | 5 (62.5) | - | - |
| TKTTF | 1 (33.3) | 3 (13.6) | - | 2 (25) | - | - |
| TKKTF | 1 (33.3) | 4 (18.2) | - | 1 (12.5) | - | 3(100) |
| TTKTF | 1 (33.3) | 4 (18.2) | - | - | - | - |
| TKPTF | - | 1 (4.5) | - | - | - | - |
| Total | 3 | 22 | 1 | 8 | 0 | 3 |

[1]Figures in parenthesis indicated the frequency (%) of a specific race in respective districts, (-) indicated the race was not present in the district

inoculated on 7-day-old seedlings 'McNair' variety, using an atomized inoculator, which does not carry known stem rust resistance genes [24].

Generally, in the Awash Basin, TTTTF was the dominant race in the Upper Awash (45.5%) and Middle Awash areas with (62.5%) in 2019/20, irrigated wheat production year. While in the Lower Awash TKKTF, was the only race affecting wheat production. We observed from these studies, *in wheat rust* epidemiology, is that if there is a race to environment interaction? Though it is difficult to conclude with two-year observations and demanded more endeavors to look at the *Pgt* race ecology, some weather data manipulation and adjustments will overcome these virulent races, in addition to gene to gene concepts of resistance breeding in Table 5.

Below is the result from CDL Minnesota (S1 Table). Also, find attached the raw data from the survey with the results of the genotyping. According to the genotyping from the Minnesota CDL lab, all nine samples were genotyped as clade III-B (Co-A04), which is associated with *Pg*t race TTRTF. These results are consistent with race phenotyping work showing that the *Pgt* races TTRTF (III-B) and TKFTF/TKKTF (IV-F) have become the prominent races in Ethiopia.

**Virulence to Sr genes.** About 100% of the races identified showed virulence above the range of 80% of the Sr genes. The races TKKTF and TKPTF showed virulence to 80% of the Sr genes. The races TKTTF and TTKTF showed 85% of virulence. The race TTTTF showed virulence to 90% of Sr genes. The race TTTTF was virulent to all differential lines except *Sr24* and *Sr31*. Races TKTTF and TTKTF differed by only one gene, which was virulent to all differential lines except *Sr11, Sr24, Sr36*, and *Sr31, Sr24, Sr36*, respectively (Table 6). The virulence pattern observed in this study conforms to the results reported by [10, 32, 33].

This study showed that the stem race races in the irrigated wheat production areas of the Awash River basin were highly virulent to most of Sr genes. The only effective genes able to provide a good harvest in the existing races are *Sr24* and *Sr31*. On the other hand, more than 80% of the Sr genes were affected, by 100% of the screened isolates. The resistance gene

**Table 6. Avirulence or virulence spectra of the *Pgt* races identified in 2018/19 and 2019/20 cool seasons irrigated wheat areas.**

| Races | Virulence | Avirulence |
|---|---|---|
| **TTTTF** | *5, 21, 9e, 7b, 11, 6, 8a, 9g, 36, 9b, 30, 17, 9a, 9d, 10, Tmp, 38, McN* | *24, 31,* |
| **TKTTF** | *5, 21, 9e, 7b, 6, 8a, 9g, 36, 9b, 30, 17, 9a, 9d, 10, Tmp, 38, McN* | *11, 24, 31* |
| **TKKTF** | *5, 21, 7b, 6, 8a, 9g, 9b, 30, 17, 9a, 9d, 10, Tmp, 9e, 38, McN* | *11, 36, 24, 31* |
| **TTKTF** | *5, 21, 9e, 7b, 11, 6, 8a, 9g,9b, 30, 17, 9a, 9d, 10, Tmp, 38, McN* | *36, 24, 31* |
| **TKPTF** | 5, 21, 9e, 7b, 6,8a,9g,36,30,17,9a,9d,10, Tmp,38, McN | 9b, 11,24,31 |

McNair (*SrMcN*) was ineffective for all isolates tested (Tables 4 and 6). During 2018/19, a total of 33 isolates were collected in the cool seasons. Out of these 33 samples, only four were viable and analyzed onto the 20 stem rust differential lines in 2018/19. From the viable samples, four isolates were identified, in the season, and four different races namely TTTTF, TKKTF, TTKTF, and TKTTF (which is also known as the Digelu race), were documented. The result was in agreement with the findings of Hei *et al*. [19], who noted that about 95% of the stem rust population was TTTTF in the rain-fed areas of Ethiopia identified from durum wheat, bread wheat, and Barley varieties during 2014–2015.

## Discussion

Irrigated Wheat crop production is grown in cool crop growing season with improved crop varieties in Awash River Basin of Ethiopia. In the basin, wheat production has been increasing to substantial levels to support wheat self-sufficiency. On the other hand, stem rust disease prevalence, incidence, and severities have been increasing and the disease may reach a level that threatens wheat production and productivity. The stem rust survey conducted over the past six years in the irrigated wheat production in the Awash River Basin of Ethiopia indicated a trend of increase in the distribution, dynamics, and physiological races of wheat stem rust with time. Recently, the rust disease intensity revealed that stem rust was prevalent in all survey districts of the Afar and Oromia regions. The increase was quite high in 2018/19 and 2019/20 production years and reached the epidemic level in Awash River Basin. The prevalence of the disease ranges from 0–100% in range with overall mean of 10%, 14.3%, 0%, 9.09%, 71.14% and 63.7% from 2014/15-2019/20, respectively. There was low stem rust prevalence in the first three years. The diseases were more prevalent in 2018 in S. Gellaalo, Amibara, Jeju, Fentale, and Sire districts indicate the disease distributed in Awash River Basin. The incidence and severity, the most important disease varied over districts, seasons, and elevations. Therefore, it is essential to take appropriate action to avert a disaster from occurring.

The intensity of the disease was increased and changed from year to year, from place to place, depending on the area's macro and microclimatic conditions. Its prevalence, incidence, and severity were increasing. This increment was alarming with the expansion of area production. The disease was also more prominent in farms near Awash River than at the distant farms from the river. The soil type and the nature of the farm also contribute to the disease. Fields surrounded by mountain ranges and clay loam soils have high pressure than the sandy loam soil types. The cool season wheat production was in lowlands, for the majority of wheat fields assessed in the Awash River Basin lays in the ranges of 360–1264 m.a.s.l., The disease was most prominent in the altitude range of (560–1264) m.a.s.l., and; its pressure was the peak in the range from (560–1064) m.a.s.l., in Gewane (50%), Fentale (35%), and Amibara (35.63%) severity in 2018/19. This result also shows the high susceptibility of the good adaptable variety Kakaba to stem rust. So, the production of resistant varieties must be supported, with the use/application of fungicide chemicals. Our results indicated that there is a wheat rust disease outbreak in the irrigated during 2018/19 and 2019/20, with the efforts for surveillance, early warning, and management options to reduce losses. The hot spot for stem rust in the irrigated production areas of Ethiopia was Upper Awash and Middle Awash areas (Figs 5 & 6).

The overall spore viability was low and very low viability recorded in 2018/19(12.12%) could be with poor handling of stem rust samples and dead spore collection in chemical sprayed fields during sampling. The overall low *Pgt*. viability could be associated with loss of sample viability with storage environmental conditions and attributed to the late arrival of the collected samples or sample miss handling. Out of the 37 viable samples analyzed, in 2018/19 and 2019/20 production seasons the races TTTTF, TKTTF, TKKTF, TTKTF, and TKPTF were

identified, by physiological race analysis. This indicated the presence of broad races with a wider virulence spectrum within the *Pgt* population in the irrigated areas. This study was in line with previous studies conducted by Hailu *et al.* [13] Hei *et al.* [19] Admassu *et al.* [33]. The pathogen race and its virulence combination varied from location to location in Awash River Basin. The variation in race composition in the study districts could be due to variations in the disease triangle, location, and time. Roelfs *et al.* [24], reported the disease prevalence of races in a specific season and region depends on the type of wheat cultivars grown and the environmental conditions, especially temperature. This area was a new production corridor of wheat, and there is an expectation of finding new races. The results showed that the races are similar to the hot season or rain-fed wheat production areas with the tendency of new race formation or introduction. Continuous wheat production makes favorable microclimate and virulent race formation, irrigation water for wheat production areas could be one of the possible reasons for the rapid evolution and high virulence diversity of the *Pgt* for the coming seasons.

The study revealed that races TTTTF and TKKTF were the most dominant in the stem rust population in Awash River Basin. The most abundant and dominant virulent race was TTTTF. It had frequencies of 25 and 45.45% of stem rust races for 2018/19 and 2019/20 irrigated wheat production areas of Afar and Oromia regions of Ethiopia. The race TTTTF had a wider virulence spectrum, also has similar virulence formula to TKTTF. The race TTTTF was virulent on 18 stem rust-resistant genes (Table 6), has a wide distribution in Upper Awash and Middle Awash areas. The race was detected in trace level in East Shewa by Lemma *et al.* [34] during 2015, TTTTF was reported to damage durum wheat, bread wheat, and barley varieties in 2014 and 2015 cropping season in rain-fed wheat production [19]. The race was also reported, in Iran during 2015 Afshari *et al.* [35] and Sicily, Italy in 2016 [4].

The second abundant and dominant virulent race was TKKTF. The race TKKTF had frequencies of 25% and 24.24% in 2018/19, 2019/20 cropping seasons, respectively. This race was the only dominant virulent race in lower awash. The TKKTF race could have a large adaptation and tolerance range for hot and lower altitude environmental factors. The resistance genes for TKKTF were *Sr11*, *Sr36*, *Sr24*, *and Sr31*. The most dominant and frequent race, in 2019 cropping season in western and southwestern Ethiopia TKKTF, containing 38.6% of isolates from the variety Danda'a, Digalu, Hidassie, Ogolcho, and local cultivars in western and southwestern Ethiopia [36].

The third abundant and virulent race was TKTTF (Digalu race). It had frequencies of 25.0 and 15.15% in 2018/19, 2019/20 cropping seasons, respectively. The third most dominant stem rust race was TKTTF also known as Digalu race virulent for 17 Sr genes (Table 6), wide distribution in Upper Awash and Middle Awash areas. The race was first detected in Ethiopia on the variety Digelu in 2012 and affected large hectares of wheat in Bale and Arsi for consecutive years [13, 37, 38] and central Ethiopia Gurage Zone [39]. The TKTTF race was also detected in Egypt in northern Africa, Iran, and Lebanon in the Middle East in 2013, 2010, and 2012 respectively [38]. The variety Digelu contains the resistant gene *SrTmp* effective against the ug99 stem race. The resistance genes *Sr31*, *Sr24*, and *Sr11* were effective against TKTTF (Table 6).

The molecular race analysis result showed one additional race called TTRTF also detected from dead samples collected from irrigated wheat fields in Fentale during 2019/20. TTRTF was also reported from Arsi and Bale in trace level by [40], the international stem rust trap nursery, and during 2015–2016 in Sakha, the most important wheat-growing region in Egypt [41]. The Race TKPTF was detected, in a single location at the Fentale district of East Shewa in 2019/20. The race TKPTF was recorded in the irrigated wheat production fields of Upper Awash in the 2019/20 Production season. Olivera *et al.* [42] reported TKPTF from Germany in 2013 summer wheat.

Races TTTTF, TKKTF, TKTTF accounted for 79.9% of the stem rust population of irrigated wheat production areas in the study period. The remaining two races were the least abundant TTKTF 18.8%, and TKPTF 1.5%. This study revealed that TTTTF was the most dominant and virulent of the stem rust populations in Oromia irrigated wheat areas. While the race, TKKTF was dominant and virulent stem rust population in the Afar region. The races were distributed, in the irrigated wheat-growing belts of Ethiopia in the study seasons. This result was in agreement with most of the recent studies in the rain-fed areas wheat production reports. As it was reported, by Lemma *et al.* [34], the virulent race of wheat stem rust commonly detected in the East Shewa Zone of central highlands of Ethiopia was TTTTF followed by TKKTF. Similar results were reported, from Iran [35] and Italy [4]. Furthermore, Yehizbalem *et al.* [43] also reported that TTTTF, TKTTF, and TKPTF, were found to be the stem rust race populations that were affecting the wheat production of North West Ethiopia.

The dominant wheat varieties like Dandaa, Fentale-2, Kakaba, and Gaambo, are adapted to irrigated wheat production areas of Awash River Basin, are susceptible to rust disease, and covered 100% of the wheat fields inspected (Table 6). This could support the development of rust epidemics in the irrigated ecologies in Ethiopia. In addition, the continued use of susceptible varieties increases the degree of a new mutant race formation to attack presently resistant cultivars. That is why the search and supply of Sr gene and varieties is continuous and eternal to avoid wheat rust epidemics. Although most of the resistant cultivars have been broken, efforts were made to develop and register resistant variety, and several, bread and durum wheat cultivars with various levels of resistance were released for production. This experience in the country emphasizes the need for genes with broader resistance or for combinations of resistance genes that can confer a broader and more durable resistance [44]. Combinations of seedling resistance with adult plant resistance in the field will provide valuable indications to select resistant varieties. The prevalence and intensity of the diseases were variable with location, distance from the river, crop type, variety, altitude range, and the growth stage of the crop. Temporal analysis indicated that a sigmoidal rise in disease levels during the wheat season and strong inter-annual variations [45]. While a simple logistic curve performs satisfactorily in predicting stem rust in some years but not outbreak patterns in other years [45].

## Summary and conclusion

Crop production in Ethiopia started to double-clutch from the extensive rain-fed dominated to intensive irrigated agriculture. However, irrigated agriculture with its merit, in changing climatic conditions, was vital for food self-sufficiency. While it needs knowledge about the full implementation of packages to production. Theoretical and practical knowledge and skills about the biotic and abiotic stresses are equally important to management in order to boom wheat productivity. For the last decade, the irrigated wheat production faced the highest stem rust incidence, and severity in Gewane and Fentale districts. While the lowest incidence and severity recorded in Lower Awash, Dubti districts. Stem rust was detected and samples were taken, at altitude ranges of (560 to 1108 m.a.s.l.). The severity of the disease was relatively high at farms nearest to Awash River banks and high dew point areas.

The disease prevalence and intensity were increasing with over years in the irrigated wheat ecology. The disease reached to epidemics level in 2018/19 and 2019/19 production years. Long-term seasonal climatic forecasts indicate temperatures are likely to be slightly above normal. These conditions will likely to favor rust development. In the new agro ecology, even though the spore pressure was little new races could exist which could coexist and leave in unknown host plants for the diseases in the ecology. New races could be expected, in unexploited potential, areas of the irrigated wheat production belts that could affect the resistant

varieties for those existing races. Most of the released varieties in irrigated wheat production areas were susceptible to rusts and care must be given to the selection of durable wheat through host resistance. The incidence and severity, of the disease, were higher on variety Fentale two than the others in research seed multiplication plots. The bread wheat variety Kakaba was selected and domesticated for its high adaptability and yield potential on all farmers' fields found susceptible to stem rust races. The highest stem rust intensity was observed in dough stage than earlier growth stages. TTTTF was among the highly virulent race identified in physiological race analysis in the area. The other economically important races identified were TKTTF (Digalu race), TKKTF, and TTKTF. Races TTRTF (genetic analysis result) and TKPTF with frequency of 1% in the study area were also the least virulent ones. Similar races were reported in the northern highlands of Gondar and Gojjam in 2010 and 2011 production years the highly virulent race TTTTF was the most dominant race which accounted for 60.4% of the races identified followed by race TKTTF with a frequency of 38.7% [46]. Differential hosts carrying *Sr24* and *Sr31* were effective genes that confer resistance to 100% of the races identified in the area. Although it is risky to use *Sr31* as the source of resistance as it is ineffective to Ug99 and corresponding variants, there were not detected on irrigated wheat belts of Awash River Basin.

Though long-distance transport is associated with the passive movement of spores in air current transmission, the disease spore induces a high invasion rate. The fast dispersal rate over a field and across fields usually linked with its polycyclic nature. So stem rust epidemics monitoring was ideal management option for this menace. Movement and invasion of virulent races like TTKSK or Ug99, and its variant gene will become virulent sporadically affects the irrigated wheat production in Ethiopia and that of the globe. New, virulent races of stem rust are a serious concern for the coming season wheat production in Ethiopia. The highly virulent Ug99 race TTKTT was increasing rapidly in rain fed wheat production areas in Guji zone. It will likely expand to the adjacent irrigated wheat belts. Many of the widely grown cultivars in the irrigated wheat areas are susceptible. An integrated network of survey and surveillance team that was functional, to conduct intensive sampling and monitoring, is recommended in cooperation with pathologists, breeders, modelers, epidemiologists, molecular geneticists, and policymakers.

## Supporting information

**S1 Table. Irrigated wheat varieties were affected by the *Pgt* races identified in 2018/19 and 2019/20 cool seasons and their frequency from the total viable *Pgt* population in the areas.** (DOCX)

**S1 Fig. This is the wheat survey plot maps in Awash Basin.** (DOCX)

**S1 Text. *Pg*t genetic diversity in upper Awash River basin of Ethiopia; core SNP assay CDL results from Minnesota.** (DOCX)

**S2 Text. Irrigated wheat field in West Arsi zone, Jeju districts midlands.** (DOCX)

## Acknowledgments

We are greatly indebted to the EIAR granting us, to conduct the survey and Werer Agricultural Research Center for the logistic supports. We are happy to thank the drivers for their

patience and willingness of my college in data collection in the warm weather. We would like to thank, Ambo wheat stem rust team Tizazu T. and Teklu N. for generating and providing physiological race analysis data. We would like also to mention thank you very much for the CDL Minnesota molecular identification teams. Last but not least, we would like to deliver my gratitude to Mr. Jemal Mohammed for his professional helping us to plot the study Maps.

## Author Contributions

**Conceptualization:** Nurhussein Seid Yesuf, Yoseph Alemayehu.

**Data curation:** Nurhussein Seid Yesuf, Sileshi Getahun, Yoseph Alemayehu.

**Formal analysis:** Nurhussein Seid Yesuf.

**Investigation:** Nurhussein Seid Yesuf, Shiferaw Hassen, Yoseph Alemayehu, Kitessa Gutu Danu, Zemedkun Alemu, Tsegaab Tesfaye, Gerald Blasch.

**Methodology:** Tsegaab Tesfaye, Netsanet Bacha Hei.

**Project administration:** Nurhussein Seid Yesuf.

**Resources:** Nurhussein Seid Yesuf, Yoseph Alemayehu, Netsanet Bacha Hei.

**Software:** Nurhussein Seid Yesuf.

**Supervision:** Nurhussein Seid Yesuf, Kitessa Gutu Danu.

**Validation:** Nurhussein Seid Yesuf.

**Visualization:** Nurhussein Seid Yesuf.

**Writing – original draft:** Nurhussein Seid Yesuf.

**Writing – review & editing:** Nurhussein Seid Yesuf, Sileshi Getahun, Yoseph Alemayehu, Gerald Blasch.

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
