## [Decision Letter · Decision Letter 0]

8 May 2021

PONE-D-21-08594

Distribution, Epidemics dynamics and physiological races of wheat stem rust (Puccinia graminis f.sp. tritici Eriks and E. Henn) on irrigated wheat in the Awash River basin of Ethiopia

PLOS ONE

Dear Dr. Yesuf,

Thank you for submitting your manuscript to PLOS ONE. After careful consideration, we feel that it has merit but does not fully meet PLOS ONE’s publication criteria as it currently stands. Therefore, we invite you to submit a revised version of the manuscript that addresses the points raised during the review process.

We look forward to receiving your revised manuscript.

Kind regards,

Yuefeng Ruan, Ph.D

Academic Editor

PLOS ONE

Journal Requirements:

5. We note that Figure 1 in your submission contain map images which may be copyrighted. All PLOS content is published under the Creative Commons Attribution License (CC BY 4.0), which means that the manuscript, images, and Supporting Information files will be freely available online, and any third party is permitted to access, download, copy, distribute, and use these materials in any way, even commercially, with proper attribution. For these reasons, we cannot publish previously copyrighted maps or satellite images created using proprietary data, such as Google software (Google Maps, Street View, and Earth). For more information, see our copyright guidelines: http://journals.plos.org/plosone/s/licenses-and-copyright.

5. 1.    You may seek permission from the original copyright holder of Figure(s) [#] to publish the content specifically under the CC BY 4.0 license. 

5.2.    If you are unable to obtain permission from the original copyright holder to publish these figures under the CC BY 4.0 license or if the copyright holder’s requirements are incompatible with the CC BY 4.0 license, please either i) remove the figure or ii) supply a replacement figure that complies with the CC BY 4.0 license. Please check copyright information on all replacement figures and update the figure caption with source information. If applicable, please specify in the figure caption text when a figure is similar but not identical to the original image and is therefore for illustrative purposes only.

6. Please ensure that you refer to Figure 2 in your text as, if accepted, production will need this reference to link the reader to the figure.

7. We note you have included a table to which you do not refer in the text of your manuscript. Please ensure that you refer to Tables 2, 3 and 5-12 in your text; if accepted, production will need this reference to link the reader to the Table.

Reviewers' comments:

Reviewer's Responses to Questions

**Comments to the Author**

1. Is the manuscript technically sound, and do the data support the conclusions?

Reviewer #1: Yes

Reviewer #2: Yes

2. Has the statistical analysis been performed appropriately and rigorously? 

Reviewer #1: Yes

Reviewer #2: I Don't Know

3. Have the authors made all data underlying the findings in their manuscript fully available?

Reviewer #1: Yes

Reviewer #2: Yes

4. Is the manuscript presented in an intelligible fashion and written in standard English?

Reviewer #1: No

Reviewer #2: Yes

5. Review Comments to the Author

Reviewer #1: Comments to authors

This paper reports a study conducted on ‘distribution, dynamics and physiological races of Pgt of wheat grown in irrigated areas of the Rift Valley region of Ethiopia’. The paper utilizes survey results of the stem rust prevalence, incidence and severity collected for five seasons. Additionally race virulence studies conducted in the lab over two seasons. I have no concern with trial set up – either for the field studies or lab work, however I have a general question whether this paper fits to PONE journal unless authors will make critical revisions and improvement in their writing. Authors used many Tables some of which can be combined and referred in the text rather. However, authors fail to refer Tables and Figures in the paper (Please below). The discussion could be improved better by discussing the research findings and removing some irrelevant topics. Please see some of my specific comments as below.

Line#14: Title ‘Epidemics’ should be written in lowercase letter as ‘epidemics’ for uniformity. Were there Pgt epidemics that happened during this study? Why are authors using this word in the title?

Abstract

Line#17: I suggest using either ‘high value’ or ‘important’ or ‘major crops’. Authors are repeating similar terms. Would this be ok? ‘Wheat (Triticum aestivum L.) is one of the highly valued cereal crops in the world.’

Line#23: ‘… and healthy plants were scored…’. How are healthy plants scored? This seems impractical and the sentence itself is vague, please reword it.

Line#26: There was no mention of location names for ‘…the two regions.’ Which regions are they representing? How about the ‘…six districts of Awash River basin’ mentioned in lines#21-22? Please make this clear.

Line#30: ‘…which, however, …’ this sentence is wordy. Please remove ‘which,..’

Line#34: Do you mean ‘These races are…?’ please replace thus with these.

Line#35: where is the evidence for this statement? TTTTF and TKKTF are the widest virulence spectrum which affects 90% of the Sr genes. Any previous research report?

ETHICAL STATEMENT

Please rewrite this: ‘Still, we give maximum care during surveying through spore-free through self-sanitation after Pgt infested field observation to minimize induced disease dissemination to the communities in the production areas that no specific permissions were required for these locations.’ It is wordy and not clear.

Introduction

Your introduction needs re-organization and condensation. For example, Paragraph 2 (lines 65-81) that presents about irrigated wheat production in Ethiopia and associated biotic and abiotic problems can be combined with paragraph 4 (lines 94-104) and shortened. Please see other paragraphs too.

Lines 55-59: The sentences are wordy and hard to follow. Please see if this is correct and replace. ‘As the world population is expected to reach nine billion by 2050 (Edmeades et al., 2010), there should be an urgent need to increase the productivity of crops like wheat to meet the increasing demand for food (Weigand, 2011). In sub-Sahara Africa (SSA) wheat is grown on a total area of 2.9 M ha with an annual production amounting 7.5 MT (FAO, 2017).’

Line#90: Leppik, (1970). Remove the comma (,) next to Leppik. Would it be ok to use more recent literatures? There is a lot of work on stem rust in East Africa following the emergence of Ug99. See literatures below and many more:

Worku Denbel, Ayele Badebo and Tameru Alemu. Evaluation of Ethiopian commercial wheat cultivars for resistance to stem rust of wheat race 'UG99'. International journal of Agronomy and Plant Production. Vol., 4 (1), 15-24, 2013 Available online at http:// www.ijappjournal.com ISSN 2051-1914 ©2013

P. D. Olivera, Y. Jin, M. Rouse, A. Badebo, T. Fetch, Jr., R. P. Singh, and A. Yahyaoui

Races of Puccinia graminis f. sp. tritici with Combined Virulence to Sr13 and Sr9e in a Field Stem Rust Screening Nursery in Ethiopia. Plant Disease 2012 96:5, 623-628

Muleta, K.T., Rouse, M.N., Rynearson, S. et al. Characterization of molecular diversity and genome-wide mapping of loci associated with resistance to stripe rust and stem rust in Ethiopian bread wheat accessions. BMC Plant Biol 17, 134 (2017). https://doi.org/10.1186/s12870-017-1082-7

M. Meyer, L. Burgin, M. C. Hort, D. P. Hodson, and C. A. Gilligan. Large-Scale Atmospheric Dispersal Simulations Identify Likely Airborne Incursion Routes of Wheat Stem Rust Into Ethiopia. Phytopathology® 2017 107:10, 1175-1186

Lines 120-121: Please be specific. Is the in outbreak of 2013 in Ethiopia? “A huge Pgt outbreak by race TKTTF in 2013 caused up to 100% yield losses in some fields (Sanders, 2011).

Line 122: the sentence is incomplete. Please rewrite.

Line 124: …national scale economic loss… this is not in your objective. Please be specific.

Line 125: …race genetic… ? I don’t understand what this means, better be replaced by …genetic analysis of isolated Pgt races in Cereals Disease Lab (CDL) at University of Minnesota … if it is ok.

Line 127: please make sure italicizing Pgt throughout the paper.

Lines 132-133: This needs rewording. Is this ok? The objective of this study was to determine the diseases intensity distribution, epidemics dynamics, in the irrigated belts of Awash River basin and to analyze the virulence of isolated Pgt races on seedling plants.

Materials and Methods

Line 145: Under sub-title ‘Survey of Wheat Stem Rust Distribution in Awash River Basin Fields’ authors did not indicate the number of fields surveyed by location and years 2014/15 to 2019/20. It is too general and unclear. I suggest authors to show how many fields in each district were visited for each growing season.

Line 159: How was ‘physiological races virulence spectrum’ assessment made? Please explain this.

Lines 172-174: ‘Five seedlings of this variety …’ it is not clear which wheat variety was used. McNair?

Line 236: Table 3 was not referred in the text.

Results

Lines 253-255: Was it only Gambo variety planted to all irrigated areas in 2016? Re-word this statement - “This means the diseases stem rust was not economically important in those seasons on wheat variety 'Gambo'. The next sentence is also self contradictory to me. If stem rust was there before 2016, how could it be new in the year 2016?

Lines 257-259: Is this ok? “In 2018/19, a total of 33 stem rust live samples were collected and 12.12% the samples were viable, whereas in 2019/20 a total of 46 samples were collected out of which 71.74% were viable.”

Line 260: ‘… for the two seasons, 46.84% of them were viable.’ Values placed in BRACKETS should be out. The next phrase and sentences should be in the discussion section. Please delete this and consider discussing later: which is low overallsamples were viable which is low overall 261 viability could be associated with loss of sample viability with storage environmental condition. The

262 low viability of samples could be attributed to the late arrival of the collected samples or sample miss

263 handling.

Lines 263-266: I can see a lack of uniformity in writing. No need to write all the values (numbers) in the text. If necessary, please use proportions rather than quantity for the ease of understanding. The details can be summarize in a Table and referred in the write up. Would this be ok? “The spatial viability the stem rust samples collected in 2019/20 in Oromia Regional State was 40% in Sire, 100% in Jeju and 70.59% in Fentale districts, while in Afar Regional State the viability was 33.3% in Afambo, 100% in Amibara and 25% in Dubti districts.

Line 275: “This could be attributed to sunny weather 15 December -15 January.” This statement is wrong. How can sunny weather increases stem rust pressure by itself. There could be several factors in the disease triangle attached to it.

Line 279: Figure 2 was not referred in the paper.

Line 304: Table 3 was not referred in the paper. Line 311: Table 4 was referred once but it was missing where it should be referred. Line 318: Table 5 was not referred in the paper. Line 320: Table 6 was not referred in the paper.

Line 324: Table 7 was not referred in the paper. Line 324: Table 8 was not referred in the paper. Line 401: Table 9 was not referred in the paper. Line 406: Table 10 was not referred in the paper. Line 410: Table 11 was not referred in the paper. Line 418: Table 12 was not referred in the paper.

Line 335: …in Figure 3A… Line 346…Figure 3B

Discussion

Line 439: …’wheat’ use lowercase letter.

Lines 439-444: The first two paragraphs are very broad or not relevant to this study.

The discussion section needs further improvement if this work is going to be published on PONE.

Line 643: Appendix Table 1 was not referred in the paper.

Reviewer #2: 1. General

- In general the manuscript was well prepared. As such findings are very important to get to the public in regard to planning approaches at developing rust resistant germplasm; it would be of great value to reflect what distinguishes this work from others, and what new findings are evident, and how might this work be used for strategies in developing better germplasm. It is also important that the statements made are as accurate as possible due to the importance of the work, I have described my comments as follows:

- Title is long

- Some areas need English editing

- Italicize gene names, eg. Page 5 (ln 223 -225), Page 6 (table 2), many places in the text

- Avoid starting sentences with acronyms

2. Abstract:

- Comments are shown on the attached document

3. Introduction

- Too much detail about wheat production in Ethiopia. Instead better if you focus on the background of the problem

- It is stated as one objective but that is not the case (Page 2, ln 132-133)

4. Materials and methods

- What percentage of incidence and/or severity rate is a considered as a cut score to classify a field as diseased during the survey?

- Figure 1 looks distorted

- At which growth stage of the plant was the survey conducted?

5. Result

- Too many tables. Move tables that are less cited in the text put them as supplemental materials.

6. Discussion

- Remove the first two paragraphs (lines 439-452) as they are less relevant. Instead, try to exploit similar works and compare with your results, ex. Meyer et al. 2021, Lemma et al. 2014, etc….

7. Conclusion

- Avoid vague conclusion (Page 19, lines 512 to 514)

6. PLOS authors have the option to publish the peer review history of their article (what does this mean?). If published, this will include your full peer review and any attached files.

Reviewer #1: **Yes: **Firdissa. E. Bokore

Reviewer #2: **Yes: **Jemanesh Haile, PhD, PAg

---

## [Author Response · Author response to Decision Letter 0]

22 Jun 2021

I accepted and acted for all comments forwarded by Plos One Editorial and Reviewers team. but i didn't work in truck change mood .

---

## [Decision Letter · Decision Letter 1]

21 Jul 2021

PONE-D-21-08594R1

Distribution, dynamics and physiological races of wheat stem rust (Puccinia graminis f.sp. tritici) on irrigated wheat in the Awash River basin of Ethiopia

PLOS ONE

Dear Dr. Yesuf,

Thank you for submitting your manuscript to PLOS ONE. After careful consideration, we feel that it has merit but does not fully meet PLOS ONE’s publication criteria as it currently stands. Therefore, we invite you to submit a revised version of the manuscript that addresses the points raised during the review process.

We look forward to receiving your revised manuscript.

Kind regards,

Yuefeng Ruan, Ph.D

Academic Editor

PLOS ONE

Journal Requirements:

Reviewers' comments:

Reviewer's Responses to Questions

**Comments to the Author**

1. If the authors have adequately addressed your comments raised in a previous round of review and you feel that this manuscript is now acceptable for publication, you may indicate that here to bypass the “Comments to the Author” section, enter your conflict of interest statement in the “Confidential to Editor” section, and submit your "Accept" recommendation.

Reviewer #1: (No Response)

Reviewer #2: All comments have been addressed

2. Is the manuscript technically sound, and do the data support the conclusions?

Reviewer #1: (No Response)

Reviewer #2: Partly

3. Has the statistical analysis been performed appropriately and rigorously? 

Reviewer #1: (No Response)

Reviewer #2: N/A

4. Have the authors made all data underlying the findings in their manuscript fully available?

Reviewer #1: (No Response)

Reviewer #2: Yes

5. Is the manuscript presented in an intelligible fashion and written in standard English?

Reviewer #1: Yes

Reviewer #2: No

6. Review Comments to the Author

Reviewer #1: Authors have made a lot of improvements over the previous draft. However, there still are some topographical errors requiring further formatting and proof reading. Below, please find a few examples:

Abstract

Line24: Sr-31 and Sr-24 could preferably written as Sr31 and Sr24. Please use similar naming in the rest of the paper.

Introduction

Line110: Please change this citation to PLOS referencing system. Please see: Kumar et al. 2011

Line 129. Author names should show up in full not as numbers at the start of any sentence. For example, [4,10] reported wheat …. should be modified as Tadesse et al. [4] and Rosegrant et al. [10].

Line 141-142: In Ethiopia, hot spot areas for the appearance of virulent genetic diversity of stem rust

races were reported by [12]. … by Hailu et al. [12]. Please make modifications throughout the paper where applicable.

Materials and methods

Line187: Please delete 2.2. from sub-tile Survey of Wheat Stem Rust Distribution in Awash River Basin Fields

Line188: The survey and surveillance assessment were assessed, among the farmers' fields at every 5-10 km ….could this be modified? My suggestion: The field assessments were conducted in the farmers’ fields at 5-10 km interval between wheat farms following the main road routes.

Line 194: Please replace Afambo&Asaita by Afambo and Asaita, otherwise it confuses readers.

Line210: via [22]… not clear. Please use author name followed by reference code. See above the comment.

Line216: Data, on disease intensity like prevalence,…. the comma (,) after ‘Data’ is not required and could be modified as... Data on disease intensity like prevalence…

Result

Line434-435: This statement doesn’t read well, please modify. Distribution and frequency of Pgt Races in Awash River basin for 2014/15 - 2017/18 off seasons in irrigated wheat cropping season there was no race analysis,

Discussion

Line498-500: This sentence has something missing or not clear! …. Did authors meant to say: The stem rust survey conducted over the past six years in the irrigated wheat production in the Awash River Basin of Ethiopia indicated a trend of increase in the distribution, dynamics, and physiological races of wheat stem rust with time.

Line594: Make a brief conclusion if any. The subtitle ‘Conclusion’ is also not necessary and could be deleted. After all, this section is seemingly a summary of the research output than a conclusion. Paragraphs placed under the conclusion can be incorporated to the discussion section.

Reviewer #2: Thank you for addressing most of the previous comments. But the manuscript still needs some cleaning and editorial work. I have included all my comments in the attached pdf file. Some of the issues are:

- Inconsistency in using acronyms

- Formatting Ex. Italicize markers and genes

- Incomplete and unclear lengthy sentences

- Conclusion part is not presented

- Unnecessary paragraph in the discussion part

7. PLOS authors have the option to publish the peer review history of their article (what does this mean?). If published, this will include your full peer review and any attached files.

Reviewer #1: **Yes: **Firdissa Bokore

Reviewer #2: **Yes: **Jemanesh K. Haile, PhD, PAg

---

## [Author Response · Author response to Decision Letter 1]

26 Jul 2021

One old reference which has been removed for the data content and the actual scenarios and facts in Ethiopia with the suggestion of reviewers. 

Response to Reviewers (2)

Once again, we would like to thank for the academic editor and reviewers of PLOS ONE journals for the nice attentive editing and comments they forwarded with in this manuscripts, it helps a lot for its improvements. Please present my gratitude for them with warm gratitude by ‘Thank you very much!” 

Saying this almost all of the comments given with the resource persons are acceptable and accepted and helps the improvement of the manuscript with the following manner. The author is not native English speaker, so there could be limitations in understanding what the editors and reviewers want to see on effect but positive to change and learn from them. Most of the reviewers’ comments are addressed. 

 

Journal Requirements:

6. Review Comments to the Author

Reviewer #1: Authors have made a lot of improvements over the previous draft. However, there still are some topographical errors requiring further formatting and proof reading. Below, please find a few examples:

Abstract

Line24: Sr-31 and Sr-24 could preferably written as Sr31 and Sr24. Please use similar naming in the rest of the paper. Accepted and I try to make similar naming in the rest of the paper 

Introduction

Line110: Please change this citation to PLOS referencing system. Please see: Kumar et al. 2011 Accepted

Line 129. Author names should show up in full not as numbers at the start of any sentence. For example, [4,10] reported wheat …. should be modified as Tadesse et al. [4] and Rosegrant et al. [10]. Accepted

Line 141-142: In Ethiopia, hot spot areas for the appearance of virulent genetic diversity of stem rust

races were reported by [12]. … by Hailu et al. [12]. Please make modifications throughout the paper where applicable. Accepted

Materials and methods

Line187: Please delete 2.2. from sub-tile Survey of Wheat Stem Rust Distribution in Awash River Basin Fields Accepted \\

Line188: The survey and surveillance assessment were assessed, among the farmers' fields at every 5-10 km ….could this be modified? My suggestion: The field assessments were conducted in the farmers’ fields at 5-10 km interval between wheat farms following the main road routes. Accepted

Line 194: Please replace Afambo&Asaita by Afambo and Asaita, otherwise it confuses readers. Accepted

Line210: via [22]… not clear. Please use author name followed by reference code. See above the comment. Acted 

Line216: Data, on disease intensity like prevalence,…. the comma (,) after ‘Data’ is not required and could be modified as... Data on disease intensity like prevalence… Acted

Result

Line434-435: This statement doesn’t read well, please modify. Distribution and frequency of Pgt Races in Awash River basin for 2014/15 - 2017/18 off seasons in irrigated wheat cropping season there was no race analysis, I try to figure out it

Discussion

Line498-500: This sentence has something missing or not clear! …. Did authors meant to say: The stem rust survey conducted over the past six years in the irrigated wheat production in the Awash River Basin of Ethiopia indicated a trend of increase in the distribution, dynamics, and physiological races of wheat stem rust with time. Exactly 

Line594: Make a brief conclusion if any. The subtitle ‘Conclusion’ is also not necessary and could be deleted. After all, this section is seemingly a summary of the research output than a conclusion. Paragraphs placed under the conclusion can be incorporated to the discussion section. 

Usually scientific studies contain conclusions from the premises which are results so could it be accepted as summary and Conclusion. Still I acted on it to improve.

Reviewer #2: Thank you for addressing most of the previous comments. But the manuscript still needs some cleaning and editorial work. I have included all my comments in the attached pdf file. Some of the issues are:

- Inconsistency in using acronyms(Accepted)

- Formatting Ex. Italicize markers and genes(Accepted)

- Incomplete and unclear lengthy sentences(Accepted)

- Conclusion part is not presented(Acted)

- Unnecessary paragraph in the discussion part(Accepted)

7. PLOS authors have the option to publish the peer review history of their article (what does this mean?). If published, this will include your full peer review and any attached files.

Yes for me!

---

## [Editor Report · Decision Letter 2]

16 Aug 2021

Distribution, dynamics and physiological races of wheat stem rust (Puccinia graminis f.sp. tritici) on irrigated wheat in the Awash River Basin of Ethiopia

PONE-D-21-08594R2

Dear Dr. Yesuf,

We’re pleased to inform you that your manuscript has been judged scientifically suitable for publication and will be formally accepted for publication once it meets all outstanding technical requirements.

Kind regards,

Yuefeng Ruan, Ph.D

Academic Editor

PLOS ONE

---

## [Editor Report · Acceptance letter]

15 Sep 2021

PONE-D-21-08594R2 

Distribution, dynamics, and physiological races of wheat stem rust (*Puccinia graminis f.sp.* tritici) on irrigated wheat in the Awash River Basin of Ethiopia 

Dear Dr. Yesuf:

I'm pleased to inform you that your manuscript has been deemed suitable for publication in PLOS ONE. Congratulations! Your manuscript is now with our production department. 

Kind regards, 

on behalf of

Dr. Yuefeng Ruan 

Academic Editor

PLOS ONE